# Pilot Study of Aerosolised Plus Intravenous Vancomycin in Mechanically Ventilated Patients with Methicillin-Resistant *Staphylococcus Aureus* Pneumonia

**DOI:** 10.3390/jcm9020476

**Published:** 2020-02-09

**Authors:** Jun Yeun Cho, Hyung-Sook Kim, Hye-Joo Yang, Yeon Joo Lee, Jong Sun Park, Ho Il Yoon, Hong Bin Kim, Jae-Joon Yim, Jae-Ho Lee, Choon-Taek Lee, Young-Jae Cho

**Affiliations:** 1Division of Pulmonary and Critical Care Medicine, Department of Internal Medicine, Chungbuk National University Hospital, Cheongju-si 28644, Korea; ok_kaist2115@hanmail.net; 2Department of Pharmacy, Seoul National University Bundang Hospital, Seongnam-si 13620, Korea; kehese2956@snubh.org; 3Division of Pulmonary and Critical Care Medicine, Department of Internal Medicine, Seoul National University Bundang Hospital, Seongnam-si 13620, Korea; yhj791019@gmail.com (H.-J.Y.); yeonjoolee1117@gmail.com (Y.J.L.); jspark.im@gmail.com (J.S.P.); dextro70@gmail.com (H.I.Y.); jhlee7@snubh.org (J.-H.L.); ctlee@snu.ac.kr (C.-T.L.); 4Division of Infectious Diseases, Department of Internal Medicine, Seoul National University Bundang Hospital, Seongnam-si 13620, Korea; hbkimmd@snubh.org; 5Division of Pulmonary and Critical Care Medicine, Department of Internal Medicine, Seoul National University Hospital, Seoul 03080, Korea; yimjj@snu.ac.kr

**Keywords:** mechanical ventilation, methicillin-resistant *staphylococcus aureus*, intensive care unit, aerosolised vancomycin, pneumonia

## Abstract

Treatment of methicillin-resistant *Staphylococcus aureus* (MRSA) pneumonia in critically ill patients remains unsatisfactory. This pilot study aimed to evaluate the clinical outcomes of aerosolised vancomycin in addition to intravenous administration in this setting. This was a prospective, noncomparative, phase II trial. Patients receiving mechanical ventilation for >48 h in intensive care units (ICUs) were screened; those receiving intravenous vancomycin for MRSA pneumonia were enrolled. Patients received aerosolised vancomycin (250 mg every 12 h for five days) via a vibrating mesh nebuliser. The primary outcome was treatment success (clinical cure or improvement) at the conclusion of antibiotic treatment. Vancomycin concentrations were measured in bronchoalveolar lavage fluid according to administration time. Twenty patients were enrolled (median age 75 years and 13 (65%) men; 18 (90%) cases with nosocomial pneumonia). Thirteen patients (65%) showed clinical cure or improvement. Microbiological eradication of MRSA was confirmed in 14 patients (70%). ICU and hospital mortality rates were 30% and 35%, respectively. Maximum aerosolised vancomycin concentration was observed 4–5 h after nebulising (98.75 ± 21.79 mcg/mL). No additional systemic adverse effects occurred following aerosol vancomycin treatment. Aerosolised vancomycin combination therapy may be an alternative treatment for patients with severe MRSA pneumonia receiving mechanical ventilation (ClinicalTrials.gov number, NCT01925066).

## 1. Introduction

Ventilator-associated pneumonia is one of the most prevalent nosocomial infections in the intensive care unit (ICU), which leads to poor treatment outcomes and high socioeconomic burden [1]. Multidrug-resistant bacteria are a common problem, and methicillin-resistant *Staphylococcus aureus* (MRSA) is a major component of Gram-positive bacteria [2].

MRSA nosocomial pneumonia is generally treated with systemic vancomycin or linezolid, but therapeutic responses are unsatisfactory [3]. Furthermore, antibiotic-associated adverse effects are problematic. Prolonged courses of systemic vancomycin treatment are frequently associated with significant nephrotoxicity [4]. Lung penetration is poor and variable, particularly in critically ill patients [5,6]. Compared with glycopeptide antibiotics, linezolid produces significantly increased risk of thrombocytopenia and gastrointestinal events [7]. 

Aerosolised antibiotics have been widely used for the treatment of multidrug-resistant nosocomial pneumonia because of its high concentration in lung tissue and favourable safety profiles [8]. A recently published guideline proposed the use of adjunctive aerosolised antibiotics in patients with nosocomial pneumonia due to multidrug-resistant Gram-negative bacilli [1]. Aerosolised vancomycin (AV) has been used as an off-label drug. One noncomparative study reported that treatment with AV plus intravenous (IV) linezolid and rifampicin for seven days achieved 95.2% clinical cure and microbiological eradication in 21 ventilated patients with MRSA pneumonia [9]. Several case reports reported that aerosolised antibiotics led to the eradication of MRSA from the lower respiratory tract [10,11,12]. However, few prospective studies have been conducted. This single-arm clinical trial aimed to evaluate the clinical outcomes of AV plus intravenous vancomycin treatment in mechanically ventilated patients with MRSA pneumonia.

## 2. Materials and Methods

### 2.1. Patients and Settings

This study was designed as a single-arm, phase II clinical trial. Adult (>19 years old) patients with MRSA pneumonia were screened for eligibility between January 2014 and December 2017 in a 1300-bed, tertiary care university-affiliated hospital. All eligible patients had to be under mechanical ventilation for more than 48 h and were undergoing IV vancomycin therapy. Number of days of IV vancomycin at time of enrolment were not considered. MRSA pneumonia was defined based on presentation of radiologic evidence of pulmonary parenchymal infection, identification of MRSA on cultures of respiratory specimens, and clinical symptoms. We excluded mixed infection cases in which Gram-negative bacteria were thought to be the dominant pathogen. Detailed inclusion and exclusion criteria are given in Appendix A.

### 2.2. Intervention

Standard IV vancomycin HCl (CJ Pharmaceuticals, Seoul, Republic of Korea) was reconstituted in 10 mL of isotonic saline. A 250 mg vancomycin solution was administered with a vibrating mesh device (Aeroneb^®®^ Pro, Aerogen Inc., Dangan, Ireland) placed proximal to the ventilator Y-connector and linked to an aerosol generator for 30 min or more, every 12 h per day. During the treatment, drug administration was not triggered by inspiration flow or pressure from the patient. The mechanical ventilator was set to constant inspiratory flow mode along with proper sedation. Aerosolised salbutamol was imperatively administered for 15 min prior to treatment to prevent unexpected bronchospasm. 

AV treatment continued for at least five days and up to seven days depending on clinical judgement. If a patient was successfully weaned from the mechanical ventilator, the study was terminated at that point. IV vancomycin dose was adjusted according to a trough level of 15–20 mcg/dL. The duration of IV treatment was determined at the discretion of the responsible physician. Protocols for AV treatment are detailed in Appendix A.

### 2.3. Vancomycin Concentration of Epithelial Lining Fluid

We obtained bronchoalveolar lavage (BAL) or trans-tracheal aspirates (TTA) to measure vancomycin concentration in the epithelial lining fluid. BAL fluid was obtained just before AV administration in patients who required bronchoscopy for other reasons 3–5 days after the start of AV treatment. BAL fluid was obtained by injecting 100 mL of normal saline in the subsegmental bronchus selected by the investigator. After 1–2 h, 4–5 h, and 11–12 h of AV administration, TTA samples were obtained. Samples were sent directly to the Department of Laboratory Medicine without preconditioning immediately after collection and measured by fluorescence polarisation immunoassay [13]. Serum drug concentration measurements and therapeutic drug monitoring for IV vancomycin treatment were performed separately according to established clinical guidelines.

### 2.4. Outcomes and Follow-Up

Primary outcome was treatment success rate (clinical cure or clinical improvement among clinical responses). Clinical responses were classified into four categories as follows: (1) clinical cure: clinical signs and symptoms were resolved relative to those at the time of diagnosis of pneumonia, a chest X-ray image showed improvement or no progression, and no further antibiotic treatment was needed; (2) clinical improvement: improvement in two or more components of clinical pulmonary infection score (CPIS) compared with the baseline [14]; (3) treatment failure: new or aggravated consolidated lesion on chest X-ray image, persistent or aggravated clinical manifestations, or death; (4) indeterminate: if the above three categories were not applicable. All cases were independently assessed by two pulmonologists (J.Y.C. and Y.J.C.) and one infectious disease specialist (H.B.K.). Judgement of treatment success was verified if three panels came to an agreement. 

Secondary outcomes were microbiologic responses, changes in CPIS, mortality (ICU and hospital), and ventilator-free days. Microbiologic responses were based on results of respiratory specimens (e.g., BAL, sputum, or TTA) and classified as follows: eradication (documented or presumed), persistence (documented or presumed), relapse, and indeterminate (i.e., missing data). Clinical and microbiologic responses were assessed at day 3 of AV treatment, end of AV treatment, and end of follow-up (EFU, defined as termination of all antibiotics). 

### 2.5. Monitoring Side Effects

In addition to the well-known side effects of vancomycin (e.g., hypotension, phlebitis, red man syndrome, drug rash or fever, dermatitis, neutropenia, and increased serum creatinine), bronchospasm or desaturation (defined when unexpected SpO_2_ decreased below 88%) were also carefully monitored during AV treatment. Patients were discontinued from the study if investigators considered these side effects to be fatal. The emergence of vancomycin-resistant strains was monitored through the hospital course based on culture results performed as needed by the attending doctor.

### 2.6. Statistical Analysis

All data are presented as mean ± standard deviation for continuous variables and numbers (percentages) for categorical variables. Data were compared between defined groups using Student’s *t*-test for continuous variables and Pearson’s chi-square test for categorical variables. Data were analysed using SPSS software for Windows version 22.0 (IBM, Armonk, NY, USA). The significance level was set at *p* < 0.05.

### 2.7. Ethical Approval and Consent to Participate

This study was approved by the Institutional Review Board and Ethics Committee of Seoul National University Bundang Hospital (Number: B-1206-156-004) and was conducted in compliance with the Declaration of Helsinki. Written informed consent was obtained from all patients or their legally authorised representative.

## 3. Results

We screened 71 patients with MRSA pneumonia who received mechanical ventilation for more than 48 h. A final total of 20 patients were enrolled in the study (Figure 1). The mean age of the patients was 74.7 ± 8.9 years, and 65% were men. MRSA bacteraemia was observed in four patients, and 13 patients (65%) had combined respiratory tract infections due to Gram-negative bacteria. During the study period, patients received an average of 15.5 days of IV vancomycin for treatment of MRSA pneumonia. Patient characteristics are detailed in Table 1.

Upon evaluating clinical responses at EFU, treatment success rate was 65% (clinical cure: *n* = 5, 25%; clinical improvement: *n* = 8, 40%) (Figure 2). CPIS was significantly decreased from the third day of AV treatment compared to baseline (7.8 ± 1.2 versus 6.0 ± 2.1, *p* < 0.05) and continuously decreased until the end of AV treatment (7.8 ± 1.2 versus 5.7 ± 2.1, *p* < 0.05) (Appendix A). Microbiologic eradication was observed in 70% of patients on the third day of AV treatment and was still observed in 70% of patients at the time of EFU without MRSA relapse or superinfection after completion of AV treatment. One patient was not followed up for microbiological responses due to transfer to another hospital.

Seven patients (35%) exhibited clinical failure or indeterminately died during hospital course (Table 2). Among them, one patient died after prolonged life-sustaining care after ICU discharge, in whom the cause of death was unclear. Two patients were successfully extubated within five days of initiating AV treatment. There were no systemic adverse events related to vancomycin treatment. Two cases of transient hypoxia were reported during AV treatment, which were immediately managed. Unexpected events included one death from rapidly developing septic shock and two cases related to airway maintenance. Vancomycin-resistant *Enterococcus* was cultured in urine and stool specimens after the completion of AV treatment in two patients, which showed colonisation. 

We obtained BAL fluid or TTA specimens of five patients for analysing vancomycin concentration. Analysis was possible in the specimens obtained from three patients. When AV was administered at intervals of 12 h, the highest concentration was observed at 4–5 h after AV administration (98.75 ± 21.79 mcg/mL). Vancomycin concentration at 11–12 h after AV administration immediately before the next administration was 8.61 ± 3.08 mcg/mL (Figure 3).

## 4. Discussion

To our knowledge, this is the first prospective study to evaluate the efficacy of additional AV treatment in mechanically ventilated patients with MRSA pneumonia, although this was a single-arm design. An additional five days of AV showed a 65% treatment success rate in patients with MRSA pneumonia who were receiving IV vancomycin treatment. Microbiologic eradication of MRSA was confirmed in 70% of patients, and CPIS was also significantly reduced in the course of treatment. These outcomes are more favourable than those reported in a previous large-scale study [15], which reported a clinical success of 46.6% and an MRSA eradication rate of 47.1%, even when considering the high severity of illness of the patients in this study.

The maximum drug concentration in the lungs (mean concentration of 98.75 mcg/mL) was observed after 4–5 h of AV administration, which is higher than that measured when using IV vancomycin alone. In a previous study which measured drug concentration in the epithelial lining fluid of 14 critically ill patients who received IV vancomycin alone for more than five days, low drug concentration was noted (mean concentration of 4.5 mcg/mL) [5]. In an experimental study in rats, drug concentrations in both lung tissue and BAL fluid via aerosol were higher than that via IV route [13]. Additional AV treatment may enable high drug concentrations in the lungs, resulting in favourable therapeutic results.

Safety profiles of AV treatment were considered acceptable. We did not observe any additional systemic adverse events that are commonly associated with vancomycin treatment. Although unexplained transient desaturation was reported during AV administration in two patients, it was not fatal; it was reversible and easily manageable. Bronchospasm, which is generally associated with inhaled antibiotic treatment, was not reported. Salbutamol pretreatment may alleviate this adverse event, which is consistent with results from a pilot study of paediatric patients with cystic fibrosis [16].

Three cases with emergence of vancomycin-resistant strains were reported. In two patients with relatively prolonged IV vancomycin (25 and 17 days, respectively), Vancomycin-resistant *Enterococcus* colonisation was observed in urine and faecal specimens. Vancomycin intermediate *Staphylococcus aureus* was incidentally detected in the respiratory specimens of one patient before the initiation of AV treatment. This patient exhibited clinical improvement after AV treatment but was not followed up after discharge, and no further culture results could be obtained. Since our study did not monitor the appearance of resistant strains through an active surveillance system, further studies are needed to determine the relationship between AV treatment and vancomycin-resistant strains.

This study has several limitations. First, the results are not generalisable due to the small sample size. Patients with underlying lung disease or other severe medical conditions were excluded because structural lung problems may confound clinical responses (i.e., especially changes in chest X-ray images) and affect the distribution of aerosol drug. Thus, strict exclusion criteria may lead to difficulty in recruiting patients. 

We performed an additional analysis to compare treatment success rates between enrolled (*n* = 17) and excluded (*n* = 17) patients, matched for age and APACHE II scores (Appendix A). The excluded group was treated with IV vancomycin alone. The enrolled group showed a higher treatment success rate than the excluded group (64.7% vs. 47.1%); however, there was no statistical significance. Future studies employing a large-scale, randomised controlled design should be conducted.

Second, 65% of the patients had Gram-negative mixed infection, and broad-spectrum antibiotics were used. As most patients had severe pneumonia, we permitted the use of broad-spectrum antibiotics determined by the physician in charge. Previously, a study on MRSA pneumonia also permitted antibiotics covering Gram-negative bacteria with the possibility of mixed infection regardless of Gram stain results [17]. In this study, quantitative culture results (>10 ^4^ CFU/mL) were obtained from BAL fluid, and antibiotics other than vancomycin were discontinued before the end of IV vancomycin in most patients. Thus, Gram-negative bacterial infection may not have had a significant effect on evaluation of clinical responses. Third, few samples were available for drug concentration analysis. Repeated bronchoscopy could not be performed taking patients’ conditions into consideration, and specimen quality may not have been suitable for drug concentration analysis. Therefore, the concentration measurement in this study should not be used as a definite reference for determining the interval of AV administration. 

In conclusion, additional AV combination therapy may be used as an alternative treatment without additional side effects for patients with severe MRSA pneumonia receiving mechanical ventilation. High drug concentration in the lung may lead to effective microbiological eradication and favourable treatment success with twice-a-day dosages. Future research on the optimal dosing schedules may be needed to support the results of the current study.

## Figures and Tables

**Figure 1 jcm-09-00476-f001:**
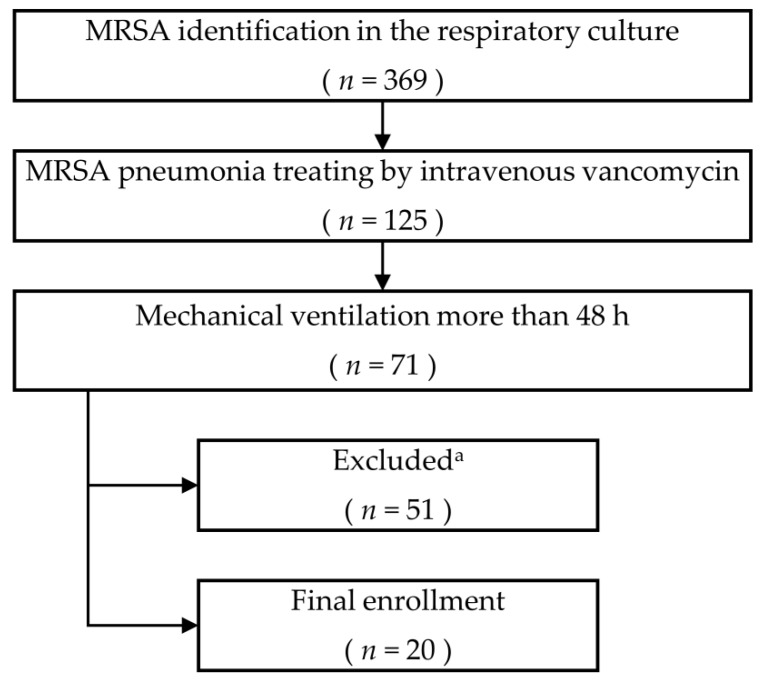
Patient recruitment. MRSA, methicillin-resistant *Staphylococcus aureus*. ^a^ Reasons for exclusion were as follows: active primary or metastatic lung cancer (*n* = 5), severe congestive heart failure (*n* = 4), severe acute respiratory distress syndrome (*n* = 4), uncontrolled asthma (*n* = 1), diffuse bronchiectasis (*n* = 2), chronic obstructive pulmonary disease (*n* = 7), combined pulmonary fibrosis and emphysema (*n* = 1), co-infection with nontuberculosis mycobacteria (*n* = 1), pleural effusion required percutaneous drainage (*n* = 3), pneumothorax (*n* = 1), destroyed lung due to previous tuberculosis (*n* = 1), viral pneumonia (*n* = 2), refusal to consent (*n* = 19).

**Figure 2 jcm-09-00476-f002:**
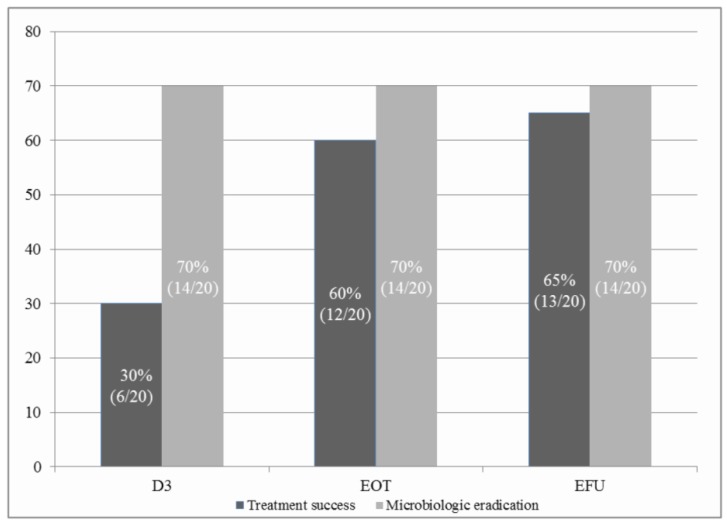
Rate of treatment success and microbiologic eradication; D3, day 3 of aerosolised vancomycin treatment; EOT, the end day of aerosolised vancomycin treatment; EFU, the end day of all types of antibiotic treatment. Treatment success includes clinical cure and clinical improvement. The ratio in the figure is defined as number of corresponded patient divided by total number of included patient.

**Figure 3 jcm-09-00476-f003:**
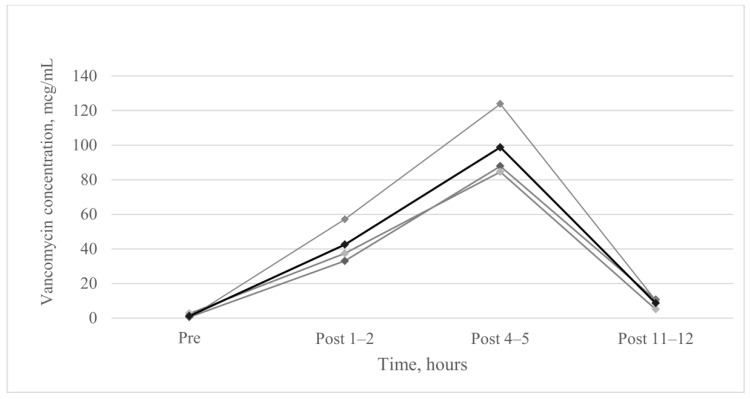
Vancomycin concentration in epithelial lining fluid. The black line denotes the average values of vancomycin concentration in epithelial lining fluid, and the grey line denotes vancomycin concentration of each of three patients. Bronchial alveolar lavage fluid or endotracheal aspirates were obtained for analysing vancomycin concentration in epithelial lining fluid. Vancomycin concentrations (mean ± standard deviation, mcg/mL) were 1.13 ± 1.27 (pre-aerosolised vancomycin), 42.49 ± 12.86 (post 1–2 h), 98.75 ± 21.79 (post 4–5 h), and 8.61 ± 3.08 (post 11–12 h), respectively.

**Table 1 jcm-09-00476-t001:** Patient characteristics.

Variables	Total (*n* = 20)
Age, years	74.7 ± 8.9
Sex, male	13 (65)
Co-morbidities	
Hypertension	9 (45)
Diabetes mellitus	6 (30)
Dyslipidaemia	2 (10)
Dementia	2 (10)
Parkinson’s disease	3 (15)
Stroke or haemorrhage	5 (25)
Ischaemic heart diseases	4 (20)
History of malignancy, not active	7 (35)
History of orthopaedic surgery	4 (20)
Pneumonia type	
Community acquired pneumonia	2 (10)
Healthcare related acquired pneumonia	3 (15)
Hospital-acquired pneumonia	11 (55)
Ventilator-associated pneumonia	4 (20)
MRSA bacteraemia	4 (20)
Gram-negative bacteria respiratory infection	13 (65)
Bronchoalveolar lavage	14 (70)
APACHE II score	21.8 ± 7.0
SOFA score	7.0 ± 3.6
CPIS^a^	7.8 ± 1.2
Median MIC, mcg/mL (range)	1.0 (0.75-1.0)
Total duration of intravenous vancomycin, days	15.5 ± 6.2
Extracorporeal membrane oxygenation	3 (15)
Renal replacement therapy	7 (35)
Tracheostomy	17 (85)

Data are presented as *n* (%) or mean ± standard deviation; MRSA, methicillin-resistant *Staphylococcus aureus*; APACHE, acute physiology and chronic health evaluation; SOFA, sequential organ failure assessment; CPIS, clinical pulmonary infection score; MIC, median inhibitory concentration; ^a^ Calculated on the variables at first time of aerosolised vancomycin treatment.

**Table 2 jcm-09-00476-t002:** Mortality, ventilator-free days, side effects, and emergence of resistant strains.

Outcomes	Total (*n* = 20)
**Mortality**	
In-intensive care unit	6 (30)
In-hospital	7 (35)
28-day	5 (25)
90-day	7 (35)
Ventilator-free day ^a^	11.0 ± 10.0
**Adverse events (any causes)**	2 (10)
Systemic	0
Localised ^b^	2 (10)
**Unexpected events**	3 (15)
Death due to Gram-negative sepsis	1 (5)
Mechanical airway obstruction	1 (5)
Accidental T-cannula dislodgement	1 (5)
**Emergence of vancomycin-resistant strains**	3 (15)
Vancomycin intermediate *Staphylococcus aureus* ^c^	1 (5)
Vancomycin-resistant *Enterococcus* ^d^	2 (10)

Data are presented as number (%) or mean ± standard deviation. ^a^ Eleven patients, except those who underwent more than 28 days of mechanical ventilation or died during mechanical ventilation. Two patients were successfully withdrawn from the mechanical ventilator before the completion of five days of aerosolised vancomycin treatment. ^b^ Desaturation events due to ventilator asynchrony. ^c^ Cultured in respiratory specimens prior to aerosolised vancomycin exposure immediately after study enrolment. ^d^ Incidentally found in urine and stool specimens after the end of aerosolised vancomycin treatment.

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
