# Peer review of "Pilot Study of Aerosolised Plus Intravenous Vancomycin in Mechanically Ventilated Patients with Methicillin-Resistant *Staphylococcus Aureus* Pneumonia"

_jcm, 2020, doi:10.3390/jcm9020476_

Round 1
Reviewer 1 Report
The need for such interventions is clear given the increased risk of MRSA poses to patients with Pneumonia where the standard of care drug (vancomycin) lacks penetration and possibly leading to poor outcomes. This study shows some human data with nebulized vancomycin. However, the lack of a comparison makes it very challenging to know the efficacy of the population. We know that there is high variation between outcomes in populations with pneumonia between institutions based on populations and capabilities. If possible to at least get a cohort with matching to compare to your intervention this would strengthen this paper significantly.
Author Response
Thank you for your good comments.
We are well aware of these shortcomings of the present study.
Due to strict inclusion criteria and low prevalence of MRSA pneumonia in our institution, we did not conduct a comparative study.
To overcome limitations of the single arm study, we compared excluded group and enrolled group matched by age and APACHE II score.
Excluded group was treated only with IV vancomycin.
Enrolled group showed higher treatment success rate than excluded group (64.7% vs 47.1%), however, there was no statistical significance.
To demonstrate the superiority of aerosol combined treatment, future studies employing a large scale, randomised controlled design should be performed.
Relevant contents have been added to Table S3 and the discussion section.
Reviewer 2 Report
Dear Author/s,
The manuscript "Pilot study of aerosolised plus intravenous vancomycin in mechanically ventilated patients with methicillin-resistant Staphylococcus aureus pneumonia by Cho et. al. is well written draft but there are some minor corrections required, which need to take care off.
1. Some of the abbreviation is need to describe before.
2. Table 1 and 2 have some subheading (Co-morbidity, Pneumonia type), format the table and make them justified and bold.
3. It will be a good and attractive, if the quality of Figures would be improvise to point of readers view.
4. Rectify the language and grammar at your end.
All the very best.
Author Response
Some of the abbreviation is need to describe before.
--> Thank you for your careful review. We modified ICU to intensive care unit (ICU) at the beginnins of the abstract. And we comfirmed that the remaining abbreviations were appropriately described.
Table 1 and 2 have some subheading (Co-morbidity, Pneumonia type), format the table and make them justified and bold.
--> We have reformatted Table 1 and 2 as your suggestions.
It will be a good and attractive, if the quality of Figures would be improvise to point of readers view.
--> Thank you for the correct point. We have inserted revised figure files with improved resolution.
Rectify the language and grammar at your end.
--> We finally corrected the language and grammar in the manuscript.
Round 2
Reviewer 1 Report
After reviewing the post-hoc analysis, it does appear that this treatment may provide a trend toward clinical benefit even if not statistically significant.
This addresses my major concern from the first draft of the paper.